# OpenAVS: Training-Free Open-Set Audio Visual Segmentation with Foundational Models

## Abstract

Audio-visual segmentation (AVS) aims to separate sounding objects from videos by predicting pixel-level masks based on audio signals. Existing methods primarily concentrate on closed-set scenarios and direct audio-visual alignment, which limits their capability to generalize to new, unseen situations. In this paper, we propose OpenAVS, a novel training-free language-based approach that, for the first time, effectively aligns audio and visual via text proxy for open-vocabulary AVS. Equipped with multimedia foundation models, OpenAVS directly infers masks through 1) audio-to-text description generation, 2) visual-to-text description generation, 3) LLM-guided prompt translation, and 4) text-to-visual sounding object segmentation. The objective of OpenAVS is to establish a simple yet flexible architecture that harnesses the strengths of appropriate foundation models, thereby maximizing their potential for effective knowledge transfer to downstream AVS tasks. Moreover, we present a model-agnostic framework OpenAVS-ST that enables the integration of OpenAVS with any advanced supervised AVS model via pseudo-label based self-training. This approach enhances performance by effectively utilizing large-scale unlabeled data when available. Comprehensive experiments on four benchmark datasets demonstrate the superior performance of OpenAVS. It surpasses existing unsupervised, zero-shot, and few-shot AVS methods by a significant margin, achieving absolute performance gains of 3.9% ∼ 6.7% and 2.2% ∼ 4.9% in mIoU and F-score, respectively, in challenging scenarios.

## 1 Introduction

Audio-Visual Segmentation (AVS) predicts dense masks of sounding objects in each video frame based on the audio signal. Despite the recent proliferation of deep AVS models, achieving state-of-the-art performance typically necessitates supervised training on a large-scale, fully-annotated dataset Gao et al. (2024); Wang et al. (2024b); Mo & Morgado (2024). While recent AVS benchmark datasets Zhou et al. (2022b; 2024); Liu et al. (2024b) have been released, models tailored to these datasets may lack generalizability, reducing their effectiveness in real-world applications where domain shifts are common.

To reduce the annotation cost, Liu *et. al.* proposed to construct a synthetic dataset by leverage existing image segmentation and audio datasets Liu et al. (2024b). Mo and Raj proposed a weakly-supervised AVS model, which generates pseudo masks based on instance-level labels Mo & Raj (2023). Unsupervised AVS methods have also been proposed to alleviate the need for ground-truth labels. For instance, Point-Prompt Yu et al. (2023) leveraged AudioCLIP Guzhov et al. (2022) to promote the Segment Anything Model (SAM) Kirillov et al. (2023) for mask generation. OWOD-BIND Bhosale et al. (2023) utilizes Open World Object Detector (OWOD) Joseph et al. (2021) to generate class-agnostic object proposals, and links them with acoustic cues using ImageBind Girdhar et al. (2023) through cosine similarities of shared latent space embeddings. MoCA Bhosale et al. (2024) adopts a self-supervised contrastive learning framework that uses DINO Caron et al. (2021) to extract visual embeddings and create positive and negative pairs.

We observe that existing unsupervised AVS methods primarily focus on exploring visual foundation models, aligning audio and visual modalities within a latent space, as illustrated in Figure 1a. These methods typically fuse audio and visual features using cross-attention Bhosale et al. (2024); Liu et al. (2024b) or similarity measures Li et al. (2022); Zhou et al. (2022a); Girdhar et al. (2023), often

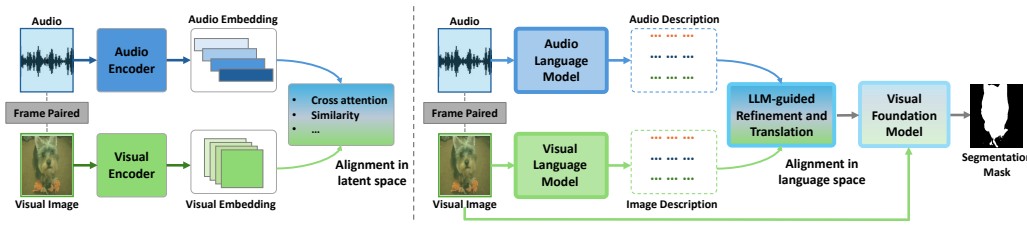

Figure 1: Different strategies for audio-visual alignment. (a) Embedding-based methods model audio-visual correlations in a latent space. (b) Language-based methods provide semantic-level alignment, enabling more effective knowledge transfer from text-audio/visual foundation models.

requiring fine-tuning to achieve better alignment of the two modalities. However, empirical studies reveal that these methods face significant challenges in complex scenarios Qian et al. (2020); Zhou et al. (2024), particularly when multiple sounding objects are present in the corresponding frame. This indicates that achieving direct alignment between audio and visual modalities is challenging possibly due to 1) limitations related to data scarcity and imbalance and 2) diverse acoustic and visual appearances of the same semantic class. Consequently, text-audio and text-visual foundation models generally demonstrate stronger generalization capabilities compared to audio-visual foundation models. Motivated by the above observations, we propose a language-based audio-visual alignment approach as shown in Figure 1b, which utilizes language as a bridge to narrow the semantic gap between audio and visual content, enabling effective knowledge transfer from audio-text, visual-text, and text-visual foundation models to fully exploit their capabilities.

Specifically, our proposed framework, termed OpenAVS, integrates off-the-shelf multi-modality (*i.e.*, audio, visual, and text) foundation models to perform unsupervised open-vocabulary AVS task. As illustrated in Figure 2, OpenAVS approaches the AVS task from a text-based perspective by decomposing it into the following four components: 1) audio-to-text description generation with Audio Language Models (ALM), 2) visual-to-text description generation with Visual Language Models (VLM), 3) LLM-guided prompt translation with Large Language Models (LLM), and 4) text-to-visual sounding object segmentation with Visual Foundation Models (VFM). By utilizing off-the-shelf ALMs such as Pengi Deshmukh et al. (2023) and Audio Flamingo Goel et al. (2025), VLMs such as Qwen-VL Bai et al. (2023); Wang et al. (2024a), and VFMs such as Grounded-SAM Ren et al. (2024b), acoustic and visual representations can be effectively aligned within the semantic space defined by LLMs such as GPT-2 Lagler et al. (2013), DeepSeek-V3 Liu et al. (2024a), and GPT-4 Achiam et al. (2023). We further identify potential gaps among ALMs, VLMs, and VFMs and propose to address this issue by jointly incorporating *Model Consistency* across ALMs and VLMs for description generation, *Prompt Consistency* across descriptions generated using different prompts, and *Frame Consistency* across consecutive frames within the same video. When unlabeled training data are available, we propose OpenAVS-ST, a model-agnostic AVS framework that improves segmentation via pseudo-label self-training, whcih outperforms existing unsupervised AVS methods on S4 and MS3 benchmarks by a significant margin.

We design our framework to be cost-efficient, achieving state-of-the-art performance with free and open-source ALMs and VLMs, while leveraging commercial LLMs such as GPT-4o-mini to enhance description quality. Notably, the cost of employing commercial LLMs in our framework remains low, as they are used solely for processing text descriptions. OpenAVS provides superior flexibility and achieves competitive or even better results compared to direct utilization of multi-modal LMs, while avoiding their drawbacks of limited model choices and substantially higher memory, resource, and API cost requirements. Here we summarize our contributions as follows:

- To the best of our knowledge, we are the first to explore ALMs for training-free AVS. The language-based approach achieves more robust audio-visual alignment than existing embedding-based methods, especially in complex multi-source scenarios.

- OpenAVS is a flexible and cost-efficient framework that links the AVS task to open-vocabulary visual tasks, enabling the utilization of advancements from a more established research domain for continuous future improvement.

- We present a model-agnostic self-training framework that seamlessly integrates OpenAVS with supervised AVS models, enhancing performance by effectively leveraging easily obtainable unlabeled data in a unified framework.
- We conduct extensive experiments to evaluate on training-free, few-shot, and zero-shot AVS tasks. Our method consistently achieves state-of-the-art performance in terms of both reliability and generalization capabilities.

## 2   RELATED WORKS

Audio-visual segmentation focuses on identifying the visual regions in a frame corresponding to the sound by generating dense pixel-level predictions. The AVSBench dataset Zhou et al. (2022b; 2024) has been widely used for training and evaluating AVS models, covering scenarios such as single sound source, multiple sound sources, and semantic segmentation. Recent expansions of AVSBench include the V3 dataset Wang et al. (2024b) for few-shot benchmarking and the AVS-Synthetic Liu et al. (2024b), which uses synthetic data to avoid manual annotations.

State-of-the-art AVS performance typically rely on supervised training Gao et al. (2024); Chen et al. (2024); Mo & Morgado (2024). A significant number of advanced fusion techniques have been proposed to learn the correlations between audio and visual modalities Gao et al. (2024); Mo & Morgado (2024). To reduce annotation cost, weakly-supervised Guo et al. (2022; 2021) and unsupervised AVS methods have also been explored to enhance model generalizability. For example, WS-AVS Mo & Raj (2023) employs weak supervision, using only class labels for class-level contrastive learning. Similarly, MoCABhosale et al. (2024) uses contrastive learning with foundational models in an unsupervised way. Direct inference methods like AT-GDINO-SAM, SAM-BIND, and OWOD-BIND Bhosale et al. (2023) integrate pre-trained models such as AST Gong et al. (2021), ImageBIND Girdhar et al. (2023), and SAM Kirillov et al. (2023). MaskCLIP+ Zhou et al. (2022a) achieves Open-Vocabulary semantic segmentation via self-training in the CLIP fashion with pseudo labels generated by DeepLab. However, existing unsupervised AVS methods are hard to generalize and usually require fine-tuning to obtain satisfactory results.

With the rise of modern foundational models, more powerful alternatives have emerged to bridge the gap between different modalities. For instance, SAM Kirillov et al. (2023) offers zero-shot segmentation and GroundingDINO Liu et al. (2025) performs open-set object detector using human inputs like category names. In addition to the aforementioned VFMs, several ALMs and VLMs, such as Pengi Deshmukh et al. (2023), Audio Flamingo Kong et al. (2024), Qwen-Audio Chu et al. (2023), LLaVA Liu et al. (2023), Qwen-VL Bai et al. (2023), and Qwen-Omni Xu et al. (2025), have been proposed recently, but they have not yet been successfully applied in existing AVS methods. Our method OpenAVS integrates ALM for the first time to address audio-visual segmentation tasks in a language-based manner, achieving a more robust and generalized audio-visual alignment.

## 3   METHODOLOGY

### 3.1   PROBLEM FORMULATION

Given paired audio signal $\mathbf{a}_i$ and visual signal $\mathbf{v}_i$ from a video, unsupervised open-vocabulary AVS aims to construct a function $\texttt{OpenAVS}_{\theta*}$ to perform audio-visual segmentation directly,

$$\mathbf{M}_i = \texttt{OpenAVS}_{\boldsymbol{\theta}*}(\mathbf{a}_i, \mathbf{v}_i) \tag{1}$$

where $\mathbf{M}_i \in \mathbb{R}^{H \times W}$ denotes the pixel-level binary mask, with $\mathbf{M}_i = 1$ indicating that the pixel belongs to a sounding object, and $\mathbf{M}_i = 0$ indicating that it belongs to the background or a silent object. We address this problem by leveraging off-the-shelf multi-modal language models and define $\boldsymbol{\theta}* \triangleq [\theta_{at}^* \mid \theta_{vt}^* \mid \theta_{tt}^* \mid \theta_{tv}^*]$ to capture audio-visual pixel-level correlations as:

$$\texttt{OpenAVS}_{\boldsymbol{\theta}*} \triangleq \texttt{VFM}_{\theta_{tv}^*}\left(\texttt{LLM}_{\theta_{tt}^*}(\texttt{ALM}_{\theta_{at}^*}, \texttt{VLM}_{\theta_{vt}^*})\right) \tag{2}$$

### 3.2   FRAMEWORK OVERVIEW

To obtain a generalized audio-visual alignment without training, we present a novel language-based alignment approach by using text as a proxy to decompose the complex AVS task into four sub-

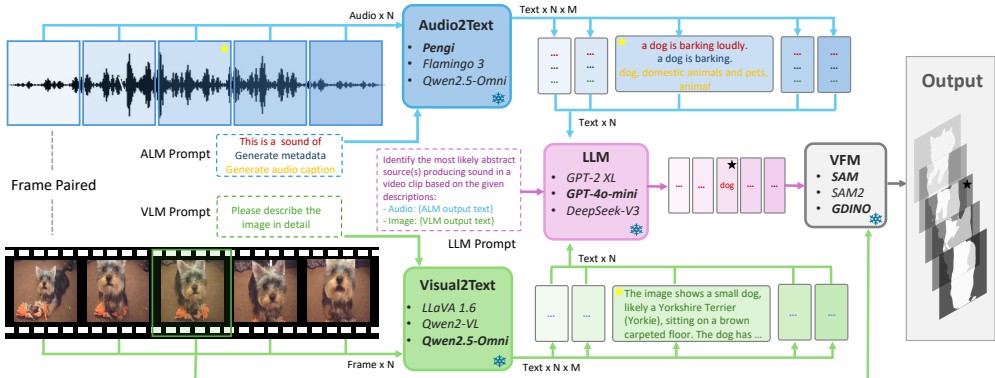

Figure 2: System overview of OpenAVS, with default models and prompt design illustrated.

tasks and solve them separately using advanced foundational models. Figure 2 depicts the overall architecture of our proposed framework, the details of which are presented in the rest of this section.

### 3.3 AUDIO-TO-TEXT DESCRIPTION GENERATION

The goal of this module is to convert the diverse audio embeddings to clear and certain semantics using an ALM. Audio embeddings extracted by the audio encoder can vary substantially, even within the same audio category or event. This variation is caused by differences in sound characteristics such as loudness and peak patterns. The varied appearances of objects in both audio and visual embeddings make it challenging to establish a direct alignment between the audio and visual feature spaces without fine-tuning. Consequently, the learned mapping is susceptible to being influenced by specific training data and application domain. To solve this issue, we employ text as a proxy to reduce the number of correlated pairs from an exponential to a linear scale.

$$\mathbf{t}_i^{(a)} = \text{ALM}(\mathbf{p}^{(a)}, \mathbf{a}_i) \tag{3}$$

As depicted in Eq.(3), an ALM takes an audio signal $\mathbf{a}_i$ and a audio-to-text prompt $\mathbf{p}^{(a)}$ as input to generate free-form text $\mathbf{t}_i^{(a)}$. With carefully designed prompt $\mathbf{p}^{(a)}$ (see Table 4), ALMs such as Pengi and Flamingo can effectively generate textual descriptions of the sounding events in the input audio.

### 3.4 VISUAL-TO-TEXT DESCRIPTION GENERATION

This module extracts text descriptions of the video frames corresponding to the audio. Let $\mathbf{p}^{(v)}$ denote the visual-to-text prompt, *e.g.*, *"Please describe the image in detail"*, and $\mathbf{v}_i$ denote the video frame. The visual description is then extracted by VLMs such as Qwen-Omni or LLaVA as:

$$\mathbf{t}_i^{(v)} = \text{VLM}(\mathbf{p}^{(v)}, \mathbf{v}_i) \tag{4}$$

We choose to generate audio and visual descriptions separately rather than directly employing a multi-modal LM due to the following reasons. First, multi-modal LMs typically require more resources, yet our empirical studies show that they offer no performance improvement over OpenAVS that adopts their single-modal counterparts. Second, OpenAVS offers greater flexibility, enabling users to configure the framework according to their performance-efficiency requirements. Comparatively, the options of multi-modal LMs are much more limited.

### 3.5 LLM-GUIDED PROMPT TRANSLATION

The audio description $\mathbf{t}_i^{(a)}$ generated by the ALM can be directly used as a prompt for VFMs such as GroundingDINO to segment the corresponding object in the image. However, this straightforward approach often fails to achieve satisfactory performance. First, the description $\mathbf{t}_i^{(a)}$ is generated solely from audio without incorporating visual information, which can lead to inaccuracies. Second,

it often contains redundant or irrelevant details, whereas the VFM favors concise subject nouns over extended expressions. One example of such issues is illustrated in Figure A1. To address the challenges, we propose utilizing an LLM, such as GPT-4o-mini, to refine and translate the audio- and visual-based descriptions, $\mathbf{t}_i^{(a)}$ and $\mathbf{t}_i^{(v)}$, into a more accurate and appropriate format as:

$$\hat{\mathbf{t}}_i^{(av)} = \text{LLM}(\mathbf{p}^{(t)}, \mathbf{t}_i^{(a)}, \mathbf{t}_i^{(v)}) \tag{5}$$

where $\mathbf{p}^{(t)}$ represents the translation prompt for the LLM. It improves the accuracy of audio descriptions by incorporating the image description $\mathbf{t}_i^{(v)}$ together with three complementary strategies:

**Model Consistency:** This strategy is applied when multiple models, including both ALMs and VLMs, are employed for description generation, ensuring that the outputs remain consistent across different models and reducing potential variability.

**Prompt Consistency:** This strategy ensures coherence across translated texts generated from distinct ALM input prompts. An LLM-based alignment translator will harmonize outputs from varied audio-to-text prompts, resolving contextual discrepancies. For example, a vague input like *"microphone"* might be refined to contextually precise terms such as *"wind, water, car engine"*.

**Frame Consistency:** This strategy enforces stability over time by aligning translations across consecutive frames from the same video source, with the assumption that audio-visual events exhibit gradual evolution rather than abrupt semantic shifts.

In summary, the LLM-guided prompt translation module effectively removes irrelevant details and emphasizes concise descriptions of the sounding object, thereby reinforcing the connection between ALM and VFM and leading to improved audio-visual segmentation performance.

### 3.6 TEXT-TO-VISUAL SOUNDING OBJECT SEGMENTATION

For training-free AVS, the raw or translated audio description, $\mathbf{t}_i^{(a)}$ or $\hat{\mathbf{t}}_i^{(av)}$, is subsequently used as a text prompt to guide VFMs in predicting segmentation masks:

$$\mathbf{M}_i = \text{VFM}(\hat{\mathbf{t}}_i^{(av)}, \mathbf{v}_i) \tag{6}$$

where $\mathbf{v}_i$ is the corresponding video frame. Specifically, OpenAVS adopts Grounded-SAM Ren et al. (2024b), which integrates GroundingDINO as an open-set object detector with the SAM as the mask predictor. This combination achieves robust segmentation performance across a wide range of visual tasks. In the experiments, we further integrated advanced vision foundation models such as SAM2 Ravi et al. (2024) and DINO-X Ren et al. (2024a) to investigate the impact of different backbone foundation models on the performance of our proposed framework.

Finally, we obtain $\mathbf{M}_i$ as the prediction result without any fine-tuning on specialized datasets. By capitalizing on the strengths of foundation models, our proposed OpenAVS demonstrates high effectiveness in the open-vocabulary AVS task for any given $(\mathbf{a}, \mathbf{v})$.

### 3.7 MODEL-AGNOSTIC AVS VIA SELF-TRAINING

To leverage the strengths of unlabeled data and supervised models, we introduce a model-agnostic framework OpenAVS-ST to seamlessly integrate OpenAVS with any existing supervised AVS models based on self-training. Instead of relying on ground-truth labels, we use the pseudo labels generated by OpenAVS as supervisory signals, while preserving the original training configurations of the adapted backbone model.

In the data preparation stage, OpenAVS process unlabeled videos or unpaired audio and image samples collected from different data sources. In either case, OpenAVS is able to effortlessly compose (image, audio, mask) triplets based on the predicted mask $\mathbf{M}_{ij}$,

$$\hat{\mathbf{Y}}_{ij} = \mathbf{M}_{ij} = \text{OpenAVS}_{\boldsymbol{\theta}^*}(\mathbf{a}_i, \mathbf{v}_j) \tag{7}$$

In the self-training stage, $\mathbf{M}_{ij}$ will be used as pseudo labels $\hat{\mathbf{Y}}_{ij}$ to optimize the following loss function, $\theta = \arg\min_\theta L(\hat{\mathbf{Y}}_{ij}, \text{AVS}_\theta(\mathbf{a}_i, \mathbf{v}_j))$ where $\text{AVS}_\theta$ refers to any supervised AVS network.

## 4 EXPERIMENTS

### 4.1 EXPERIMENT SETUP

**Dataset** We evaluate our proposed method on four benchmark datasets, namely **S4**, **MS3**, **AVSS**, and **V3** from AVSBench. Following previous work Wang et al. (2024b); Liu et al. (2024b), we convert the semantic labels of **AVSS** and **V3** to object labels by $\mathbf{Y}_i^{(object)} = \min(\mathbf{Y}_i^{(semantic)}, \mathbf{1})$.

**Evaluation Metric** We adopt the mean Intersection over Union (mIoU) and F-score to evaluate our model. A higher mIoU value implies better region similarity, while a higher F-score indicates improved contour accuracy.

**Implementation Details** We implemented three variants of OpenAVS with different configurations:

- **OpenAVS-Lite**: Pengi as the ALM without a VLM component, and GPT-4o-mini as the LLM that refines $\mathbf{a}_i$ only by considering both *Prompt Consistency* and *Frame Consistency*.
- **OpenAVS**: Pengi as the ALM, Qwen2.5-Omni as the VLM, and GPT-4o-mini as the LLM that jointly refines $\mathbf{a}_i$ and $\mathbf{v}_i$ by considering both *Model Consistency* and *Prompt Consistency*.
- **OpenAVS-Large**: Pengi, Audio Flamingo 3, and Qwen2.5-Omni as ALMs, Qwen2.5-Omni as the VLM, and GPT-4o-mini as the LLM that jointly refines $\mathbf{a}_i$ and $\mathbf{v}_i$ by incorporating *Model Consistency* and *Prompt Consistency*.

The LLM prompt designs for different variants are provided in Appendix F. For text-to-visual sounding object segmentation, we use GroundingDINO 1.0 for detection (box threshold = 0.25) and SAM (default) or SAM2 as the segmentation VFM across all variants. Experiments were conducted on a single Tesla V100 32GB GPU machine, with Xeon CPU and 64GB RAM.

### 4.2 COMPARISON WITH THE STATE-OF-THE-ARTS

As shown in Table 1, we compare our proposed OpenAVS to the 17 state-of-the-art unsupervised AVS methods, including 7 open-set approaches: Point-Prompt and Box-Prompt Yu et al. (2023), AT-GDINO-SAM, SAM-BIND, and OWOD-BIND Bhosale et al. (2023), OV-AVSS Guo et al. (2024), and AL-Ref-SAM2 Huang et al. (2025).

#### 4.2.1 RESULTS ON OPEN-SET UNSUPERVISED AVS

Training-free open-set AVS approaches use foundation models for open-vocabulary audio-visual segmentation directly on test samples, without data collection or model training. As shown in Tables 1a, OpenAVS outperforms existing methods by a significant margin on MS3 and AVSS datasets, where multiple sounding objects are present in the frames. By replacing the default SAM in OpenAVS-Lite with SAM2 (OpenAVS-Lite (SAM2)), we achieve a significant performance boost, demonstrating the framework's ability to evolve and improve in step with advances in foundation models. On the S4 dataset, AL-Ref-SAM2 demonstrates stronger performance. It leverages the strong vision-language capabilities of SAM2 and GPT-4-turbo through a two-step refinement process, which incurs higher costs due to image-based inputs. In contrast, our method relies on GPT-4o-mini with a single-shot text query, resulting in significantly lower cost (see Table 3). This also demonstrates the flexibility of our approach for performance-cost tradeoffs.

#### 4.2.2 RESULTS ON TRAINING-BASED UNSUPERVISED AVS

Training-based unsupervised methods use an unlabeled dataset to learn audio-visual correlations through self-supervised learning. Although such methods typically achieve better results within the same domain, they lack open-set capabilities, resulting in degraded performance when applied to unseen scenarios. Recall that we extend OpenAVS within a self-training framework to leverage unlabeled data when available. It offers improved performance on seen classes in the training dataset, but compromises the ability to perform open-set segmentation. To reflect this trade-off, we refer to it as OpenAVS-ST, where ST denotes self-training. The experimental results demonstrate the superior performance of our proposed method. By simply using the generate pseudo labels as supervision signals, our method is able to outperform recent, carefully designed training-based approaches for the AVS task on both the S4 and MS3 datasets, which verifies the effectiveness of OpenAVS-ST.

Table 1: Comparison with open-set and training-based unsupervised AVS methods on S4, MS3, AVSS datasets. TF and OS refer to Training-Free and Open-Set, respectively.

(a) Open-Set AVS.

| Method | TF | OS | S4 | | MS3 | | AVSS-Binary | |
|---|---|---|---|---|---|---|---|---|
| | | | mIoU | F-score | mIoU | F-score | mIoU | F-score |
| AT-GDINO-SAM | ✓ | ✓ | 0.380 | 0.460 | 0.250 | 0.290 | - | - |
| Point-Prompt | ✓ | ✓ | 0.403 | 0.515 | 0.288 | 0.333 | - | - |
| SAM-BIND | ✓ | ✓ | 0.420 | 0.510 | 0.280 | 0.360 | - | - |
| Box-Prompt | ✓ | ✓ | 0.512 | 0.615 | 0.418 | 0.478 | - | - |
| OWOD-BIND | ✓ | ✓ | 0.580 | 0.670 | 0.340 | 0.440 | - | - |
| OV-AVSS (USSL) | ✗ | ✓ | 0.486 | 0.616 | 0.361 | 0.427 | 0.525 | 0.617 |
| AL-Ref-SAM2 | ✓ | ✓ | **0.705** | **0.811** | 0.486 | 0.535 | 0.592 | 0.662 |
| OpenAVS-Lite | ✓ | ✓ | 0.582 | 0.689 | 0.483 | 0.565 | 0.593 | 0.654 |
| OpenAVS-Lite (SAM2) | ✓ | ✓ | 0.638 | 0.728 | 0.512 | **0.587** | 0.617 | 0.667 |
| OpenAVS (SAM2) | ✓ | ✓ | 0.680 | 0.764 | 0.511 | 0.541 | 0.651 | 0.704 |
| OpenAVS-Large (SAM2) | ✓ | ✓ | 0.684 | 0.769 | **0.525** | 0.557 | **0.659** | **0.711** |

(b) Training-based Unsupervised AVS.

| Method | TF | OS | S4 | | MS3 | |
|---|---|---|---|---|---|---|
| | | | mIoU | F-score | mIoU | F-score |
| WS-AVS Mo & Raj (2023) | ✗ | ✗ | 0.341 | 0.518 | 0.309 | 0.469 |
| LVS Chen et al. (2021) | ✗ | ✗ | 0.379 | 0.510 | 0.295 | 0.330 |
| Mix-Localize Hu et al. (2022) | ✗ | ✗ | 0.440 | 0.690 | 0.320 | 0.360 |
| MSSL Qian et al. (2020) | ✗ | ✗ | 0.449 | 0.663 | 0.261 | 0.363 |
| EZ-VSL Mo & Morgado (2022) | ✗ | ✗ | 0.450 | 0.680 | 0.280 | 0.340 |
| 3DC Mahadevan et al. (2020) | ✗ | ✗ | 0.571 | 0.759 | 0.369 | 0.503 |
| AGL-SSL Park et al. (2024) | ✗ | ✗ | 0.598 | 0.690 | 0.411 | 0.467 |
| iGAN Mao et al. (2025) | ✗ | ✗ | 0.616 | 0.778 | 0.429 | 0.544 |
| SST Duke et al. (2021) | ✗ | ✗ | 0.663 | 0.801 | 0.426 | 0.572 |
| MoCA Bhosale et al. (2024) | ✗ | ✗ | 0.680 | 0.790 | 0.570 | 0.620 |
| OpenAVS-Lite-ST | ✗ | ✓ | 0.693 | 0.824 | 0.556 | 0.649 |
| OpenAVS-Lite-ST (SAM2) | ✗ | ✓ | 0.703 | 0.823 | 0.561 | 0.663 |
| OpenAVS-ST (SAM2) | ✗ | ✓ | 0.719 | 0.834 | **0.582** | 0.661 |
| OpenAVS-Large-ST (SAM2) | ✗ | ✓ | **0.732** | **0.845** | 0.578 | **0.674** |

Table 2: Comparison with few-shot and zero-shot AVS methods on the V3 dataset.

| Method | 0-shot | | 1-shot | | 3-shot | | 5-shot | |
|---|---|---|---|---|---|---|---|---|
| | mIoU | F-score | mIoU | F-score | mIoU | F-score | mIoU | F-score |
| SAM-Fusion | 0.463 | 0.630 | 0.504 | 0.671 | 0.571 | 0.719 | 0.608 | 0.741 |
| TPAVI | 0.530 | 0.707 | 0.561 | 0.754 | 0.632 | 0.767 | 0.639 | 0.783 |
| AVSegFormer | 0.543 | 0.715 | 0.583 | 0.764 | 0.642 | **0.774** | 0.652 | 0.785 |
| GAVS | 0.547 | 0.722 | 0.629 | **0.768** | 0.663 | **0.774** | 0.678 | **0.795** |
| OpenAVS-Lite | 0.663 | 0.736 | 0.692 | 0.755 | 0.695 | 0.760 | 0.696 | 0.761 |
| OpenAVS-Lite (SAM2) | **0.675** | **0.741** | **0.699** | 0.757 | **0.702** | 0.761 | **0.703** | 0.762 |

### 4.2.3 RESULTS ON FEW-SHOT AND ZERO-SHOT AVS

For few-shot and zero-shot AVS, we compare our proposed OpenAVS to SOTA methods, SAM-Fusion Wang et al. (2024b), TPAVI Zhou et al. (2022b), AVSegFormer Gao et al. (2024), and GAVS Wang et al. (2024b) in Table 2 on V3 dataset Wang et al. (2024b). Our training-free OpenAVS significantly outperforms existing approaches in mIoU, with absolute improvements of 12.8% in zero-shot setting, 7.0%, 3.9%, and 2.5% in 1-shot, 3-shot, and 5-shot scenarios. Additionally, without using any training samples, our method achieves the highest F-score in the zero-shot setting, while all competing methods are trained on seen data from V3. Thus, the results verify the robustness of our proposed method to unseen scenarios in an open-set setting.

Table 3: Performance and cost comparison of LLMs on S4 dataset.

| Model | LLM | VFM | mIoU | F-score | Cost/video |
|---|---|---|---|---|---|
| OpenAVS-Lite | GPT-2 XL | GDINO+SAM | 0.431 | 0.561 | $0 |
| OpenAVS-Lite | DeepSeek-V3 | GDINO+SAM | 0.576 | 0.681 | 0.00125 CNY |
| OpenAVS-Lite | GPT-4o-mini | GDINO+SAM | 0.582 | 0.689 | $0.000154 |
| OpenAVS-Lite | GPT-4o-mini | GDINO+SAM2 | 0.638 | 0.728 | $0.000154 |
| OpenAVS | GPT-4o-mini | GDINO+SAM2 | 0.680 | 0.764 | $0.00135 |
| OpenAVS-Large | GPT-4o-mini | GDINO+SAM2 | 0.684 | 0.769 | $0.00163 |

Table 4: Ablation study of the audio-to-text description generation and LLM-guided description translation modules in OpenAVS-Lite on S4 dataset.

| Audio-to-text prompt | ALM | #Para | w/o LLM | | w/ GPT-4o-mini | |
|---|---|---|---|---|---|---|
| | | | mIoU | F-score | mIoU | F-score |
| *This is a sound of* | Pengi | 0.3B | 0.549 | 0.635 | 0.568 | 0.657 |
| *Generate metadata* | Pengi | 0.3B | 0.551 | 0.640 | 0.564 | 0.653 |
| *Generate audio caption* | Pengi | 0.3B | **0.560** | 0.650 | **0.567** | 0.656 |
| *Please describe the audio in detail* | Audio Flamingo 3 | 8.2B | 0.558 | **0.672** | 0.551 | 0.684 |
| *Please describe the audio in detail* | Qwen2.5-Omni-7B | 10.7B | 0.548 | 0.663 | 0.563 | **0.692** |

## 4.3 ABLATION STUDY

### 4.3.1 PERFORMANCE–COST ANALYSIS WITH VARYING LLMS

We compare performance and cost with varying LLMs and report the results in Table 3. For OpenAVS-Lite, the transition from GPT-2 XL to DeepSeek-V3 and GPT-4o-mini leads to significant improvements in both mIoU and F-score, with only a marginal increase in cost. Notably, GPT-4o-mini paired with GDINO+SAM2 achieves the best performance while keeping the cost per video as low as $0.000154, demonstrating that high-quality segmentation can be achieved without relying on expensive models. For OpenAVS and OpenAVS-Large, the image descriptions extracted by the VLM are much longer than the audio descriptions extracted by the ALM. As a result, the number of prompt tokens for the LLM to process increases substantially, leading to enhanced performance but higher unit cost for both OpenAVS and OpenAVS-Large.

### 4.3.2 AUDIO-TO-TEXT DESCRIPTION GENERATION

ALMs take an audio recording and a text prompt as input to generate audio descriptions as output. Here, we evaluate the impact of using different ALMs with varying audio-to-text prompts to generate the free-form text, and we present the results in Table 4. The results indicate that using *"Generate audio caption"* to prompt Pengi is generally more robust for AVS tasks. For OpenAVS and OpenAVS-Large, the LLM-based translator is required to combine both audio and visual descriptions. Comparatively, for OpenAVS-Lite, where only the audio description is available, the translator is optional. These experiments show that the LLM-based translator not only aligns audio and visual descriptions but also bridges the gap between the ALM's output and the VLM's input to achieve enhanced performance.

### 4.3.3 MODEL EFFICIENCY ANALYSIS WITH VARYING ALMS AND VLMS

Table 5 reports the performance–efficiency analysis of our models with SAM2 as the segmentation VFM. For methods with multiple processing steps, the reported time cost is given as a range: the lower bound corresponds to fully parallel execution, while the upper bound reflects fully serialized processing. For instance, Pengi (with all three prompts) requires 0.541 s under serial execution, but only 0.180 s with parallel execution, yielding roughly a threefold speedup. Compared with Flamingo and Qwen, Pengi is more lightweight (see Table 4) and more efficient. VLMs incur substantially higher time costs than ALMs, as images are more complex and produce more tokens than audio. Additionally, the LLM (GPT-4o-mini) requires an average of 0.729 s, while the segmentation VFM (GDINO+SAM/SAM2) takes around 0.311 s. As a result, the end-to-end pipeline latency ranges

Table 5: Model efficiency analysis with varying ALMs and VLMs on the S4 dataset. (1xGPU 32GB)

| Model | ALM | VLM | mIoU | F-score | Time (sec/frame) |
|---|---|---|---|---|---|
| OpenAVS-Lite | Pengi | - | 0.630 | 0.718 | 0.180 |
| OpenAVS-Lite | Pengi (all three prompts) | - | 0.639 | 0.725 | 0.180 - 0.541 |
| OpenAVS-Lite | Audio Flamingo 3 | - | 0.592 | 0.684 | 0.788 |
| OpenAVS-Lite | Qwen2.5-Omni-7B | - | 0.604 | 0.695 | 1.192 |
| OpenAVS | Pengi | Qwen2.5-Omni-7B | 0.678 | 0.762 | 4.090 - 5.080 |
| OpenAVS | Pengi (all three prompts) | Qwen2.5-Omni-7B | 0.680 | 0.764 | 4.090 - 5.441 |
| OpenAVS | Qwen2.5-Omni-7B | Qwen2.5-Omni-7B | 0.665 | 0.752 | 4.090 - 6.092 |
| OpenAVS-Large | All Three ALMs | Qwen2.5-Omni-7B | 0.684 | 0.769 | 4.090 - 7.421 |
| - | Qwen2.5-Omni-7B (multi-modal) | | 0.662 | 0.752 | 5.669 |

| Raw Image | Ground Truth | AVSS (USSL) | OV-AVSS | OpenAVS (CL)[a] | OpenAVS (PO)[b] | OpenAVS |
|---|---|---|---|---|---|---|

**Clip-level class label**: tabla    **Pengi**: a drum loop is being played.    **OpenAVS**: drum

**Clip-level class label**: baby_woman    **Pengi**: a woman is saying something and a man is saying something.    **OpenAVS**: a woman

[a] Use Clip-level class label text    [b] Use raw Pengi output text

Figure 3: Visual comparison of OpenAVS variants vs. baselines on challenging AVS cases.

from 1.22 to 2.23 s for OpenAVS-Lite and from 5.13 to 6.709 s for OpenAVS with the VLM component included. Recall that all experiments were conducted on a single Tesla V100 32GB GPU machine. With more powerful hardware (*e.g.*, H200), the processing time could be further reduced.

Finally, we compare with the multi-modal Qwen2.5-Omni. It directly consumes both audio and visual inputs, along with the prompt "*Please describe what you hear based on what you see.*" However, its performance falls short in both effectiveness and efficiency compared to our method.

### 4.3.4 VISUALIZATIONS

Figure 3 illustrates the performance comparison between OpenAVS and other approaches on MS3. OpenAVS shows strong robustness, even in challenging scenarios where objects of the same category (*e.g.*, a baby and a woman) appear in the same frame. The results also highlight that OpenAVS can handle a variety of text prompt inputs, which significantly influence the performance. Using Pengi outputs directly (OpenAVS (PO)) exhibits instability in producing masks, as the raw outputs are not well-suited for VFMs. This underscores the importance and effectiveness of our LLM-based translator, which bridges the modality gap between foundational models.

## 5 CONCLUSION

We explore the open-vocabulary audio-visual segmentation problem, which generates pixel-level masks of the sounding objects in each frame by leveraging the superior generalization capability of foundation models. A new open-set and training-free framework OpenAVS has been presented in this paper, which innovatively connects audio and visual foundation models through language, enabling effective knowledge transfer from both text-audio and text-visual foundation models. OpenAVS achieves state-of-the-art performance on three benchmark datasets, surpassing existing unsupervised methods and showcasing strong generalizability to unseen domains.

## REPRODUCIBILITY STATEMENT

We have taken steps to ensure the reproducibility of our work. All model architectures, training details, and hyper-parameter settings are described in the main text and appendix. Complete experimental protocols and dataset preprocessing procedures are provided in the supplementary materials. We will release the source code and scripts for data processing and evaluation in the camera-ready version to further facilitate replication.

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

## A  LLM USAGE

We clarify that LLM (ChatGPT) was used only for polishing the grammar and readability of the manuscript during the paper writing. The LLM was not involved in generating ideas, methods, analyses, or results. All scientific content is entirely the authors' original work.

## B  DATASETS AND EVALUATION METRICS

### B.1  DATASET DETAILS

We evaluate our method on the following benchmark datasets that vary in complexity, scale, and semantic diversity.

- **S4** consists of 4,932 video clips covering 23 distinct classes, each containing a single sounding object. This dataset serves as a standard benchmark for evaluating segmentation performance in clean and well-separated audio-visual scenarios.

- **MS3** contains 424 samples from the same 23 classes as S4 but features multiple concurrent sounding sources per clip, making it more challenging due to overlapping audio events and increased ambiguity in visual cues.

- **AVSS (V2)** extends the **S4** and **MS3** datasets scale and diversity, featuring 12,356 videos across 70 categories. It includes upgraded versions of the original 5,356 videos and 7,000 newly collected multi-source clips.

- **V3** includes 11,356 video clips spanning 70 object categories, formed by merging **MS3** with the **V2** dataset Zhou et al. (2024). It introduces a larger and more diverse label space and is split into seen and unseen categories to facilitate evaluation under few-shot and zero-shot settings Wang et al. (2024b). This makes V3 particularly suitable for assessing the open-vocabulary generalization capabilities of audio-visual segmentation methods.

Together, these datasets provide a comprehensive testbed for evaluating performance across both standard and open-set scenarios, covering a range of challenges from single-source to multi-sources audio-visual events and from limited to large-scale category diversity.

### B.2  EVALUATION METRICS

The evaluation metrics used in this paper are defined as follows.

**Mean of Intersection over Union (mIoU).** For binary audio-visual segmentation task, the IoU for the foreground class is given by:

$$IoU = \frac{TP}{TP + FP + FN} \tag{8}$$

where $TP$, $FP$, and $FN$ denote true positives, false positives, and false negatives, respectively.

The mean IoU (mIoU) is then defined as:

$$mIoU = \frac{1}{2}\left(IoU_{\text{fg}} + IoU_{\text{bg}}\right) \tag{9}$$

where $IoU_{\text{fg}}$ corresponds to the segmented object (foreground), and $IoU_{\text{bg}}$ corresponds to the background.

**F-score.** The generalized $\mathcal{F}_\beta$ score is defined as:

$$\mathcal{F}_\beta = \frac{(1 + \beta^2) \times \text{Precision} \times \text{Recall}}{\beta^2 \times \text{Precision} + \text{Recall}} \tag{10}$$

where we set $\beta^2 = 0.3$, following Zhou et al. (2022b).

Table A1: Comparison with supervised and unsupervised methods on AVSS dataset. GT refers to ground-truth labels.

| Method | TF | GT | mIoU | F-score |
|---|---|---|---|---|
| Audio-SAM | ✗ | ✓ | 0.574 | 0.684 |
| SAM-Fusion | ✗ | ✓ | 0.602 | 0.724 |
| TPAVI | ✗ | ✓ | 0.625 | 0.756 |
| GAVS | ✗ | ✓ | 0.677 | 0.788 |
| OV-AVSS (USSL) | ✗ | ✗ | 0.525 | 0.617 |
| AL-Ref-SAM2 | ✓ | ✗ | 0.592 | 0.662 |
| OpenAVS | ✓ | ✗ | 0.593 | 0.654 |
| OpenAVS-Lite (SAM2) | ✓ | ✗ | 0.617 | 0.667 |
| OpenAVS-Large (SAM2) | ✓ | ✗ | 0.659 | 0.711 |

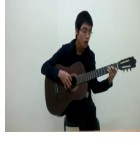 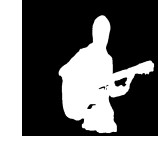 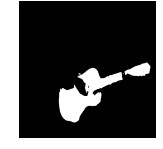 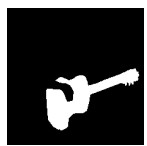

(a) Raw frame     (b) *A person is playing a guitar*     (c) *Refined output: Guitar*     (d) GT mask

Figure A1: Translate audio description from (b) to (c) to improve segmentation accuracy.

## C ADDITIONAL EXPERIMENTAL RESULTS

### C.1 EXTENDED RESULTS ON AVSS

On the binarized AVSS dataset, we compare our method not only with unsupervised approaches, such as OV-AVSS Guo et al. (2024) and AL-Ref-SAM2 Huang et al. (2025) (Table 1), but also with supervised methods that leverage ground-truth masks, including Audio-SAM, SAM-Fusion, GAVS Wang et al. (2024b), and TPAVI Zhou et al. (2022b) in Table A1. The results demonstrate that our unsupervised, training-free OpenAVS not only outperforms other unsupervised competitors but also achieves performance competitive with these supervised methods.

### C.2 EXAMPLE OF LLM-GUIDED PROMPT TRANSLATION

As described in Section 3.5, OpenAVS employs LLM-guided prompt translation to refine the output of the Audio Language Model (ALM), mitigating issues caused by misleading or ambiguous descriptions. For instance, as shown in Figure A1, although a person appears in the frame, he was not producing any sound. Using the raw ALM output $\mathbf{t}_i^{(a)}$ directly could mislead the Visual Foundation Model (VFM) into segmenting irrelevant objects since it will capture the word *"person"* as well.

The results in Table 4 verify the effectiveness of the LLM-guided prompt translator, with performance gains observed across all cases when enhanced by LLM (GPT-4o-mini). Moreover, Table A2 shows that both **prompt** and **frame** consistency strategies are capable of addressing aforementioned potential issues. While frame consistency may not yield as significant an improvement as prompt consistency, it did not require any additional ALM API calls hence is meaningful for balancing cost, inference time, and performance in real-world applications.

### C.3 IMPACT OF USING CLIP-LEVEL WEAK LABELS

We further investigate the effect of replacing the LLM-translated prompts with clip-level weak labels, defining a "soft" upper bound of our OpenAVS-Lite, denoted as OpenAVS-Lite*. This approach is inspired by WS-AVS Mo & Raj (2023), which leverages clip-level annotations instead of pixel-level masks to reduce annotation costs in AVS tasks.

As shown in Table A3, using weak labels provides only marginal improvements on the S4 and MS3 datasets. However, performance still falls short of our OpenAVS (SAM2) and OpenAVS-Large

Table A2: Ablation study of LLM translation strategies

| Prompt Consistency | Frame Consistency | S4 | | #API calls per video |
|---|---|---|---|---|
| | | mIoU | F-score | |
| ✗ | ✗ | 0.567 | 0.656 | 5 |
| ✗ | ✓ | 0.569 | 0.674 | 5 |
| ✓ | ✗ | 0.581 | 0.687 | 15 |
| ✓ | ✓ | **0.582** | **0.689** | 15 |

Table A3: Impact of using clip-level text labels (OpenAVS-Lite*)

| Method | S4 | | MS3 | | V3 (0-shot) | |
|---|---|---|---|---|---|---|
| | mIoU | F-score | mIoU | F-score | mIoU | F-score |
| OpenAVS-Lite | 0.582 | 0.689 | 0.483 | 0.565 | 0.663 | 0.736 |
| OpenAVS-Lite (SAM2) | 0.638 | 0.728 | 0.512 | 0.587 | 0.675 | 0.741 |
| OpenAVS-Lite* | 0.648 | 0.748 | 0.536 | 0.600 | 0.656 | 0.726 |

(SAM2) results reported in Table 1. One contributing factor is that clip-level labels apply to the entire audio clip, including silent periods, which can lead to inaccurate segmentation since they fail to capture temporal variations in the scene.

## D  MODEL-AGNOSTIC AVS VIA SELF-TRAINING

### D.1  SUPERVISED AVS MODELS

To support model-agnostic AVS via self-training, we test three segmentation models with the following setups:

- **TPAVI**: uses `PVTv2-B5` and `VGGish`; trained on 1 GPU (32G) for 15 epochs, batch size 4, learning rate 1e-4.

- **AVSegformer**: uses `PVTv2-B5` and `VGGish`; trained on 2 GPUs (32G) for 30 epochs, batch size 2, learning rate 2e-5.

- **SAMA-AVS**: uses `SAM` (`sam_vit_h`) and `VGGish`; trained on 2 GPUs (32G) for 80 epochs, batch size 2, learning rate 2e-4.

### D.2  HYPER-PARAMETER SETTING

One of the key hyperparameters that influences the quality of generated pseudo labels for self-training or zero-shot inferencing is the box threshold used in GDINO. To determine an appropriate value, we conducted experiments as shown in Figure A2. For direct inference tasks, the optimal box threshold falls between 0.25 and 0.35 for both the S4 and MS3 datasets. A lower threshold tends to reduce accuracy by including irrelevant regions, while a higher threshold dramatically decreases the number of mask pixels, negatively impacting performance also.

In contrast, our self-training OpenAVS-Lite-ST is not sensitive to this hyperparameter, as pseudo labels generated using box thresholds ranging from 0.25 to 0.75 yield similar performance. This is attributed to the model's auto-correction capability, which compensates for either higher-quality but smaller pseudo label sets or lower-quality but more extensive ones. Based on these observations, we adopt a box threshold of 0.25 for all experiments by default.

### D.3  SELF-TRAINING AVS WITH DIFFERENT BACKBONES

We perform comprehensive experiments to evaluate our proposed self-training framework, OpenAVS-ST, with different supervised backbones: TPAVI Zhou et al. (2022b), AVSegFormer Gao et al. (2024), and SAMA-AVS Liu et al. (2024b). The results are presented in Table A4. The results show that GPT-4o-mini substantially boosts mIoU and F-score across multiple AVS backbones. This

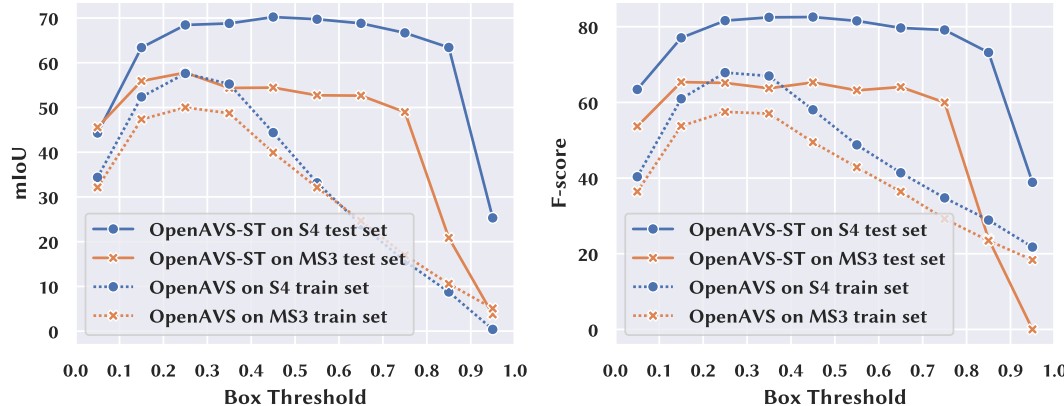

Figure A2: Impact of box threshold variation on segmentation performance (mIoU and F-score) for S4 and MS3. The "-Lite" suffix is omitted from the legend labels for clarity.

Table A4: Enhancement of text prompt via LLM and VFM upgrade, using OpenAVS-Lite-ST and OpenAVS-Large-ST with SOTA backbones.

| AVS Model | VLM[a] | LLM[b] | VFM[c] | S4 | | MS3 | |
|---|---|---|---|---|---|---|---|
| | | | | mIoU | F-score | mIoU | F-score |
| TPAVI | ✗ | ✗ | GM1 | 57.63 | 69.86 | 36.58 | 46.40 |
| | ✗ | ✓ | GM1 | 59.19 (↑1.6) | 71.60 (↑1.7) | 40.22 (↑3.6) | 50.83 (↑4.4) |
| | ✗ | ✓ | GM2 | 63.52 (↑5.9) | 75.75 (↑5.9) | 43.08 (↑6.5) | 51.85 (↑5.5) |
| AVSegFormer | ✗ | ✗ | GM1 | 66.81 | 78.73 | 47.23 | 57.62 |
| | ✗ | ✓ | GM1 | 68.21 (↑1.4) | 80.06 (↑1.3) | 51.92 (↑4.7) | 61.70 (↑4.1) |
| | ✗ | ✓ | GM2 | 70.30 (↑3.5) | 82.30 (↑3.6) | 53.28 (↑6.1) | 61.60 (↑4.0) |
| | ✓ | ✓ | GM2 | 73.15 (↑6.3) | 84.47 (↑5.7) | 52.27 (↑5.0) | 63.29 (↑5.7) |
| SAMA-AVS | ✗ | ✗ | GM1 | 60.72 | 70.81 | 53.47 | 62.24 |
| | ✗ | ✓ | GM1 | 65.64 (↑4.9) | 76.54 (↑5.7) | 55.37 (↑1.9) | 64.26 (↑2.0) |
| | ✗ | ✓ | GM2 | 69.35 (↑8.6) | 79.16 (↑8.4) | 56.10 (↑2.6) | 66.29 (↑4.1) |
| | ✓ | ✓ | GM2 | 69.02 (↑8.3) | 79.29 (↑8.5) | 57.82 (↑4.4) | 67.39 (↑5.2) |

[a] ✓indicates that the VLM (Qwen2.5-Omni) is enabled, while ✗indicates not
[b] ✓indicates that the LLM (GPT-4o-mini) is enabled, while ✗indicates not
[c] GM1 = GDINO+SAM, GM2 = GDINO+SAM2

highlights the effectiveness of text-based enhancement in bridging ALMs and VFMs, particularly when the models lack consistent textual training. Incorporating additional text descriptions extracted by VLMs from the visual input can further improve the results. Moreover, upgrading from SAM to SAM2 enhances self-training results, yielding improvements of approximately 5.5% in mIoU and 5.3% in F-score. Notably, SAMA-AVS achieves the best prediction results on the multi-source dataset **MS3**. This is because SAMA-AVS is built on SAM's frozen backbone, which performs outstandingly in segmentation tasks with limited data.

### D.4 LIMITATION AND IMPACT OF FOUNDATION MODELS

Although OpenAVS is limited by existing foundation models, its performance will improve as these models advance. For example, as shown in Figure A3, the current GroundingDINO model (DINO 1.0) fails to ground the box in a case where the image presents an uncommon view of a horse, making it a challenging recognition task. The upgraded commercial version, DINO-X, achieves remarkable performance improvements, highlighting its strong potential for real-world applications. Furthermore, upgrading from SAM to SAM2 also leads to substantial improvements, as shown earlier.

| (a) Raw Image | (b) GT Mask | (c) DINO 1.0 | (d) DINO 1.0 + SAM | (e) DINO-X | (f) DINO-X + SAM |

Figure A3: GDINO: *"horse walking on a hard surface"*.

# E  ADDITIONAL VISUAL ILLUSTRATIONS

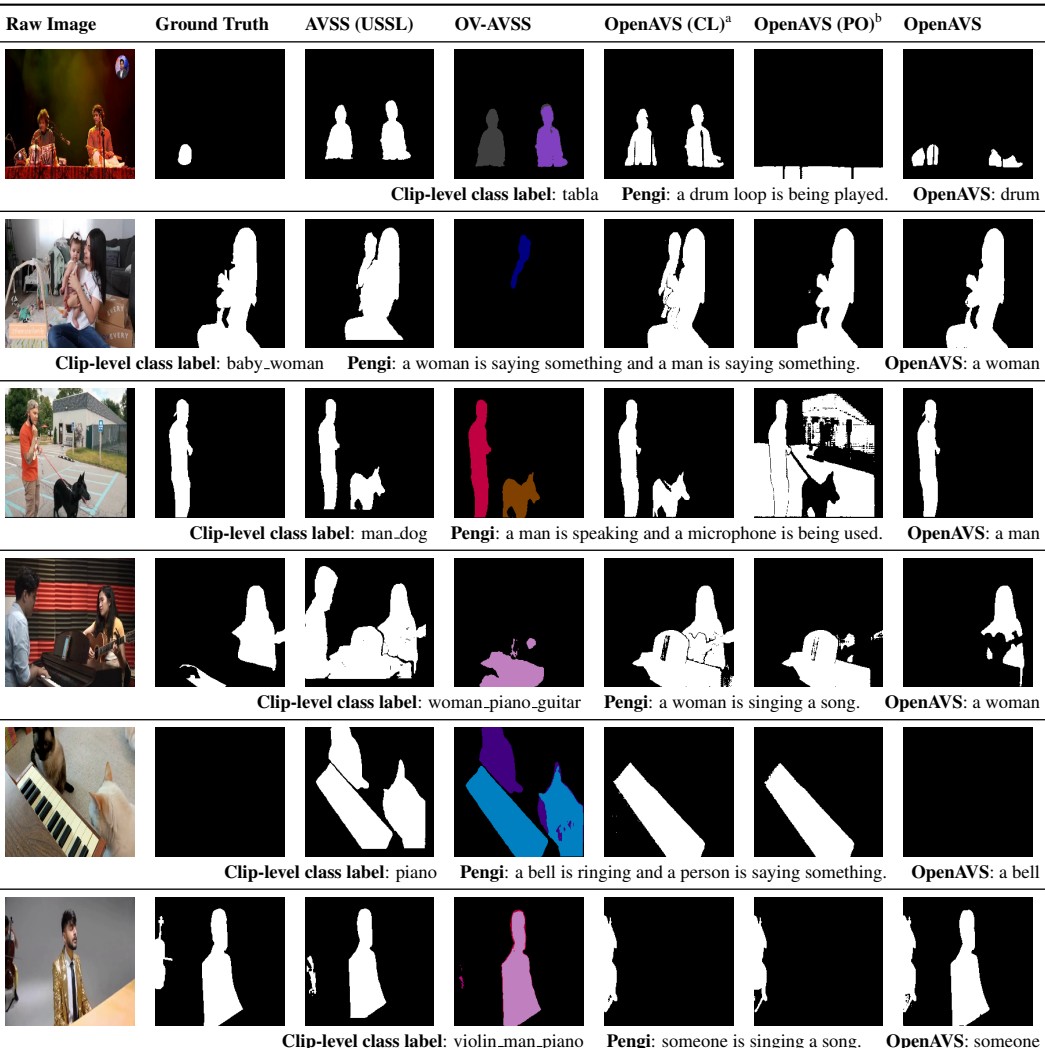

Figure A4: Visual comparison of OpenAVS variants vs. baselines on challenging AVS cases.

[a] Use Clip-level class label text    [b] Use raw Pengi output text

To supplement the main results presented in the original Figure 3, we provide additional qualitative examples in Figure A4 to further illustrate the effectiveness and robustness of OpenAVS under challenging audio-visual conditions. These examples, omitted from the main paper due to space constraints, follow the same evaluation setup.

The visual comparisons showcase various OpenAVS variants alongside baseline methods, emphasizing cases with ambiguous sound sources, overlapping audio events, and visually complex scenes.

Figure A5: Illustration of Prompt and Frame Consistency in the LLM-Based Translator for Improving ALM Outputs.

OpenAVS consistently generates more accurate and temporally coherent segmentation masks by leveraging language-guided open-set inference. These extended results support the findings reported in the main paper and offer deeper insight into the generalization capabilities of our approach.

It is worth noting that method OV-AVSS was originally designed for semantic segmentation tasks and follows a two-stage pipeline: (a) Universal Sound Source Localization (USSL) and (b) Open-Vocabulary Classification (OVC). The first stage module USSL is trained to locate all objects in an image given the corresponding audio signal. Subsequently, the second stage module OVC leverages holistic class-level label text to semantically filter the localized objects using CLIP Radford et al. (2021). Since our work does not incorporate semantic information, we compare against the first-stage module of their approach, referred to as OV-AVSS (USSL) in Table 1.

## F  PROMPTS AND INPUTS FOR LLM TRANSLATION STRATEGIES

As described in Section 3.5, we designed 3 consistency strategies, namely **prompt consistency**, **frame consistency**, and **model consistency**, to enhance the translation quality produced by the LLM-based prompt translator. An illustration of these strategies is shown in Figure 2 and A5.

For each 1-second video segment, its audio is fed into the ALM (Pengi), which takes a fixed text prompt and generates a description of what it "hears" in that segment. In our setup, we use three fixed prompts: *"This is a sound of"*, *"Generate metadata"*, and *"Generate audio caption"*, as shown in Figure A5. Additionally, we use general prompt *"Please describe the audio in detail"* for other ALMs like Audio Flamingo and Qwen2.5-Omni, and *"Please describe the image in detail"* for VLM like Qwen2.5-Omni. These prompts can be interpreted as simulating different expert perspectives on the same audio input. Each prompt produces one output per segment (or frame), so a 5-second video results in 5 frames of output per prompt.

To better leverage both **temporal continuity** across frames and **semantic diversity** across prompts, the LLM-based translator applies three forms of consistency:

- **Frame consistency**, which encourages alignment across consecutive time frames.

- **Prompt consistency**, which enforces agreement among the outputs generated by different prompts for the same frame.

- **Model consistency**, achieved by ensembling the text description outputs of different ALMs and VLMs to enhance quality from multiple perspectives.

The details of consistency prompting and the format of user input are presented in the following sections.

### F.1 BASIC TRANSLATOR

**System prompt**:

> **System Prompt for basic Translator**
>
> ```
> You are participating in a competitive game where your goal
> is to identify the most likely abstract source(s) (e.g.,
> human, instrumental, etc.)  that is/are producing sound in
> a given audio clip.  This clip was broken down into several
> frames, each containing multiple audio outputs generated by
> different AIs, representing sounds at a specific timestamp.
> Each frame corresponds to a different moment in the same
> video clip and some frames may contain no sound-producing
> objects at all, or the text output could provide misleading
> information.
>
> Your task:
> - Identify and output only the object(s) producing sound in
> each frame.
> - For each frame, provide your guess in one line, (seperate
> by comma if multiple objects), enclosed in with <answer> and
> </answer> tag pair.
> ```

**User input**:

> **User Input for Basic Translator**
>
> ```
> <frame0>
>   wind is blowing and a car engine is running
> </frame0>
> <frame1>
>   rain is falling and the wind is blowing
> </frame1>
> <frame2>
>   a motor is running and a car engine is revving.
> </frame2>
> <frame3>
>   a car engine is revving up and revving down.
> </frame3>
> <frame4>
>   a car engine is running and a car engine is running.
> </frame4>
> ```

### F.2 TRANSLATOR WITH PROMPT CONSISTENCY

**System prompt**:

> **System Prompt for Translator with Prompt Consistency**
>
> ```
> You are participating in a competitive game where your goal
> is to identify the most likely abstract source(s) (e.g.,
> human, instrumental, etc.)  that is/are producing sound in
> a given audio clip.  This clip was broken down into several
> frames, each containing multiple audio outputs generated by
> different AIs, representing sounds at a specific timestamp.
> ```

```
Each frame corresponds to a different moment in the same
video clip and some frames may contain no sound-producing
objects at all, or the text output could provide misleading
information.

Your task:
- Analyze the outputs from all audio AIs in each frame
together.
- Identify and output only the object(s) producing sound in
each frame.
- For each frame, provide your guess in one line, (seperate
by comma if multiple objects), enclosed in with <answer> and
</answer> tag pair.
```

**User input**:

**User Input for Translator with Prompt Consistency**

```
<frame0>
  <exp1>wind is blowing and a car engine is running </exp1>
  <exp2>a stream of water is flowing.  </exp2>
  <exp3>the sounds of water, wind and wind noise (microphone)
</exp3>
</frame0>
<frame1>
 <exp1>rain is falling and the wind is blowing.  </exp1>
  <exp2>a sound is being recorded.  </exp2>
 <exp3>wind </exp3>
</frame1>
<frame2>
 <exp1>a motor is running and a car engine is revving.
</exp1>
 <exp2>engine2.  sound of an engine </exp2>
 <exp3>engine </exp3>
</frame2>
<frame3>
 <exp1>a car engine is revving up and revving down.  </exp1>
 <exp2>engine sound.  i recorded a sound from a car engine.
</exp2>
 <exp3>engine </exp3>
</frame3>
<frame4>
  <exp1>a car engine is running and a car engine is running.
</exp1>
  <exp2>a sound is being played.  </exp2>
  <exp3>engine </exp3>
</frame4>
```

F.3    TRANSLATOR WITH FRAME CONSISTENCY

**System prompt**:

**System Prompt for Translator with Frame Consistency**

```
You are participating in a competitive game where your goal
is to identify the most likely abstract source(s) (e.g.,
human, instrumental, etc.)  that is/are producing sound in
a given audio clip.  This clip was broken down into several
frames, each containing multiple audio outputs generated by
different AIs, representing sounds at a specific timestamp.
Each frame corresponds to a different moment in the same
video clip and some frames may contain no sound-producing
objects at all, or the text output could provide misleading
information.

Your task:
- Consider the relationships among frames.
- Identify and output only the object(s) producing sound in
each frame.
- For each frame, provide your guess in one line, (seperate
by comma if multiple objects), enclosed in with <answer> and
</answer> tag pair.
```

**User input**:

**User Input for Translator with Frame Consistency**

```
<frame0>
 wind is blowing and a car engine is running
</frame0>
<frame1>
 rain is falling and the wind is blowing
</frame1>
<frame2>
 a motor is running and a car engine is revving.
</frame2>
<frame3>
  a car engine is revving up and revving down.
</frame3>
<frame4>
 a car engine is running and a car engine is running.
</frame4>
```

## F.4 TRANSLATOR WITH BOTH PROMPT AND FRAME CONSISTENCY

**System prompt**:

**System Prompt for Translator with Prompt and Frame Consistency**

```
You are participating in a competitive game where your goal
is to identify the most likely abstract source(s) (e.g.,
human, instrumental, etc.)  that is/are producing sound in
a given audio clip.  This clip was broken down into several
frames, each containing multiple audio outputs generated by
different AIs, representing sounds at a specific timestamp.
Each frame corresponds to a different moment in the same
video clip and some frames may contain no sound-producing
```

```
objects at all, or the text output could provide misleading
information.

Your task:
- Analyze the outputs from all audio AIs in each frame
together.
- Consider the relationships among frames.
- Identify and output only the object(s) producing sound in
each frame.
- For each frame, provide your guess in one line, (seperate
by comma if multiple objects), enclosed in with <answer> and
</answer> tag pair.
```

**User input**:

**User Input for Translator with Prompt and Frame Consistency**

```
<frame0>
 <exp1>wind is blowing and a car engine is running </exp1>
 <exp2>a stream of water is flowing.  </exp2>
 <exp3>the sounds of water, wind and wind noise (microphone)
</exp3>
</frame0>
<frame1>
 <exp1>rain is falling and the wind is blowing.  </exp1>
 <exp2>a sound is being recorded.  </exp2>
 <exp3>wind </exp3>
</frame1>
<frame2>
 <exp1>a motor is running and a car engine is revving.
</exp1>
 <exp2>engine2.  sound of an engine </exp2>
 <exp3>engine </exp3>
</frame2>
<frame3>
  <exp1>a car engine is revving up and revving down.  </exp1>
 <exp2>engine sound.  i recorded a sound from a car engine.
</exp2>
 <exp3>engine </exp3>
</frame3>
<frame4>
 <exp1>a car engine is running and a car engine is running.
</exp1>
  <exp2>a sound is being played.  </exp2>
 <exp3>engine </exp3>
</frame4>
```

## F.5 TRANSLATOR WITH MODEL CONSISTENCY

**System prompt**:

**System Prompt for Translator with Model Consistency**

```
You are participating in a competitive game:  identify
the most likely abstract source(s) (e.g., human, animal,
```

```
instrumental, mechanical) producing sound in a video clip -
based only on textual descriptions.

You are given:

- Multiple image descriptions (Image 0, Image 1, ...).  Each
is a frame caption or visual summary generated by a separate
agent; they do NOT share information.

- Multiple audio descriptions (Audio 0, Audio 1, ...).  Each
describes what the sound is approximately like (e.g., "sounds
like a motorcycle idling") and is generated by a separate
agent; they do NOT share information.

Your required procedure:

- Extract visual evidence:  For each image description,
identify and list the explicit or clearly implied objects.

- Extract acoustic evidence:  For each audio description,
identify the key acoustic cues.

- Within- and cross-modality synthesis.

- From all audio agents, compare and consolidate the cues
into an overall audio profile.  This synthesized audio
profile does not need to be a verbatim phrase from the given
descriptions; it should capture the best generalization of
the sound.

- Final decision:  Use the synthesized audio profile and
visual profile to decide which objects are most likely
producing the sound.

Output:

- First give a clear, concise, step-by-step reasoning
that references description labels (e.g., "Image 2 shows
a lawnmower; Audio 1 describes a low rumble similar to
lawnmower idling - supports lawnmower").

- After that reasoning, output the final decision on a single
line only, listing the object(s) most likely producing the
sound separated by commas when necessary.

- Enclose the single-line final answer in `<answer>` and
`</answer>` tags and place nothing else on that line.
```

**User input**:

```
User Input for Translator with Model Consistency

Image agent 0:  The image shows a person with long dark hair
wearing a white sleeveless top with a black pattern.  They
are are holding a green parrot on their shoulder.  The parrot
has has a light-colored beak and is perched calmly on the
```

person's hand.  In the background, there is a black metal
cage with a white cushion inside, and some other of the room
is visible, including a white wall and a dark-colored object
that appears to be a piece of furniture or a shelf.  The
overall setting seems to be indoors, possibly in a living
room or a similar space.

Audio agent 0:  The audio contains a single word spoken by a
female voice in a neutral tone.  The word is 'yes'.

Audio agent 1:  a person is walking on a carpet.  someone is
making a sound.  this audio contains sound events:  clothing,
domestic sounds and home sounds.

Audio agent 2:  The audio contains a sound event that is
being described.

