# OpenReview forum: "OpenAVS: Training-Free Open-Set Audio Visual Segmentation with Foundational Models"
_ICLR.cc/2026/Conference — ICLR 2026 Conference Withdrawn Submission_

### Official Review · Reviewer_2d29 · 2025-10-16

**Soundness:** 3
**Presentation:** 3
**Contribution:** 2
**Rating:** 4
**Confidence:** 4

**Summary:**

The authors propose OpenAVS, a novel training-free language-based approach that, for the first time, effectively aligns audio and visual via a text proxy for open-vocabulary AVS. But there is still work to do.

**Strengths:**

1. The method seems feasible.
2. The writing is clear and easy to read.
3. Enough visualization is provided.

**Weaknesses:**

1. The citation style appears inconsistent with ICLR guidelines. Mixing \citet{} and \citep{} hinders readability and should be standardized.

2. Integrating multiple mature systems introduces clear drawbacks. Please include a runtime comparison for each model in Table 3, and analyze potential error accumulation across the pipeline.

3. The paper evaluates on S4, MS3, AVSS, and V3 from AVSBench, which are typically used in closed-set settings. Even though the method is training-free, comparisons against closed-set trained baselines are still important. Moreover, evaluation on an open-vocabulary AVS dataset would be more appropriate, given that LLMs operate in an open-vocabulary regime.

4. Evaluation details are missing. Please report the inference resolution and the average input/output token counts for the LLM.

5. You state “GroundingDINO 1.0 for detection (box threshold = 0.25).” How critical is the box threshold? Provide an ablation or sensitivity analysis to justify the chosen value.

6. In Figure 3, AVSS is a classification task where every mask should be assigned a color. Why do the GT and other model outputs lack color assignments? Please clarify the visualization protocol or correct the figure.

Extra: I think this training-free method based on LLMs is naturally suitable for Ref-AVS [2] tasks. Why not test on Ref-AVS tasks

[1] Guo, R., Qu, L., Niu, D., Qi, Y., Yue, W., Shi, J., ... & Ying, X. (2024, October). Open-vocabulary audio-visual semantic segmentation. In Proceedings of the 32nd ACM International Conference on Multimedia (pp. 7533-7541).

[2] Wang, Y., Sun, P., Zhou, D., Li, G., Zhang, H., & Hu, D. (2024, September). Ref-avs: Refer and segment objects in audio-visual scenes. In European Conference on Computer Vision (pp. 196-213). Cham: Springer Nature Switzerland.

**Questions:**

My main concern is the evaluation and analysis of the overall system. I will finalize my score after the rebuttal.

---

### Official Review · Reviewer_CRVn · 2025-10-30

**Soundness:** 2
**Presentation:** 3
**Contribution:** 1
**Rating:** 2
**Confidence:** 5

**Summary:**

The manuscript proposes a training-free, open-vocabulary audio-visual segmentation (AVS) framework, OpenAVS, which uses language as a mediator to alleviate cross-modal misalignment in multi-source and temporally drifting scenarios. The method first converts audio and visual frames into semantic text via audio language models (ALMs) and vision-language models (VLMs), then employs a large language model (LLM) to translate and consolidate prompts with model/prompt/frame consistency constraints. The refined noun-centric prompts guide visual foundation models (VFMs) (e.g., Grounded-SAM/SAM2) to produce pixel-level masks frame-by-frame. TThe method attains competitive results on several benchmarks.

**Strengths:**

1). The paper is generally well written and easy to follow, with a clear problem setup and pipeline description.

2). The method shows reasonable performance without task-specific training in the reported settings.

3). The pipeline is modular and can be instantiated with alternative ALMs/VLMs or VFMs with minimal changes.

4). By operating in text space, the approach can accommodate unseen categories to a limited extent, offering broader semantic coverage than fixed-class models.

**Weaknesses:**

1). Incremental contribution. The framework largely assembles off-the-shelf components, and similar workflow-style pipelines have appeared in prior work [1–3], including AVSS/AVVS systems that convert audio into textual/context cues to guide frame selection [1].

2). Limited novelty of ALM usage. The headline contribution—introducing an ALM to obtain text from audio—seems straightforward. Related LLM/agent literature [4, 5] already invokes specialized modules coordinate tasks, which makes the methodological advance here appear modest.

3). Multi-source disambiguation is insufficiently analyzed, particularly for overlapping or co-occurring sound sources and attribution when two similar sources are active.

4). The approach may be sensitive to prompt templates and LLM choices, and the paper provides limited ablations on prompt variants and threshold settings.

5). Latency concerns. The reported inference time (≈5.13–6.71 s for the best settings) suggests the method is far from practical deployment. For typical videos at 25–35 FPS, real-time operation would require ~28–40 ms per frame; the current latency is orders of magnitude higher unless substantial optimization is shown.

6). Questionable cost–effectiveness. The best configuration (e.g., OpenAVS-Large + GPT-4o-mini + GDINO+SAM2) attains mIoU 0.684 / F 0.769 at ~$0.00163 per sample, yet still trails task-specific trained baselines. It is unclear why one should pay per-inference costs for comparatively lower accuracy. Conversely, lighter variants (e.g., OpenAVS-Lite + GPT-2 XL + GDINO+SAM at mIoU 0.431 / F 0.561) fall below thresholds that would be usable in practice.

7). Metric concerns: the reviewer highlights that Jaccard and IoU differ in segmentation evaluation. I refer to the SAM2 paper to emphasize this distinction and note that previous methods were computed using Jaccard.


[1] Unleashing the temporal-spatial reasoning capacity of gpt for training-free audio and language referenced video object segmentation.

[2] Open-Vocabulary Audio-Visual Semantic Segmentation

[3] Retrieval-Augmented Generation for AI-Generated Content: A Survey

[4] Agentic Reasoning: A Streamlined Framework for Enhancing LLM Reasoning with Agentic Tools

[5] Mind2web: Towards a generalist agent for the web

**Questions:**

nil.

---

### Official Review · Reviewer_sshA · 2025-10-31

**Soundness:** 3
**Presentation:** 3
**Contribution:** 2
**Rating:** 4
**Confidence:** 5

**Summary:**

This paper introduce a novel training-free language-based approach for open-set audio visual segmentation.
The language-based method achieves more robust audio-visual alignment than existing methods.
This is a flexible and cost-efficient framework, and achieves state-of-the-art performance on four benchmark and training-free, few-shot, and zero-shot AVS tasks.

**Strengths:**

* This is  a flexible, model-agnostic and cost-efficient framework.
* Experiments on 4 benchmarks show the good performance of OpenAVS.
* It surpasses existing unsupervised, zero-shot, and few-shot AVS methods.

**Weaknesses:**

* I believe this method is likely to work effectively, as none of the steps appear problematic. However, it seems more focused on the engineering aspect rather than offering new insights to the community.

* Are there any specific challenges or difficulties in applying this pipeline to solve the task?

* Given the large number of models used, is it reasonable to combine so many for this task, especially in comparison to other approaches?

* I am unclear about how few-shot AVS is applied in the proposed method. The pipeline appears primarily focused on generating masks from audio signals. Where are the few-shot examples integrated into this process?

* More details on the self-training process would be helpful. For example, which specific parts of the model are being trained during this phase?

* Could you clarify the role of the V2T component? It might also be beneficial to conduct ablation studies to better understand the contributions of both the A2T and V2T components.

I would be happy to revise my score if the author addresses these points.

**Questions:**

Please refer to the weakness.

---

### Official Review · Reviewer_k19a · 2025-10-31

**Soundness:** 2
**Presentation:** 3
**Contribution:** 2
**Rating:** 4
**Confidence:** 5

**Summary:**

This paper focuses on audio-visual segmentation, i.e., separating sounding objects from videos by predicting pixel-level masks of audio signals. Instead of direct audio-text alignment, this paper use one training-free idea, namely OpenAVS, to aligns audio and visual via text proxy. And the pipeline is divided into audio-to-text description generation, visual-to-text description generation, LLM-guided prompt translation, and text-to-visual sounding object segmentation. Experiments are carried out on four benchmarks, across unsupervised, zero-shot, and few-shot AVS settings.

**Strengths:**

[+] The manuscript is well written, with clear logics and sufficient formulations.

[+] Exploring the omni-modal alignment of image-text-audio is one promising direction.

[+] Experiments are conducted across unsupervised, zero-shot, and few-shot AVS settings.

**Weaknesses:**

[-] The bottleneck of information compression. Text is a highly compressed form of audio/images, which means a lot of information is lost when converting audio/image to text. Many textual descriptions are coarse-grained, for example, audio-to-text usually cannot capture differences between dog breeds. If a video contains two different breeds of dogs, OpenAVS may fail after converting audio/image to text.

[-] The noise is gradually amplified. This paper’s idea is severely limited by the performance of the audio/image to text pre-trained models. What should be done if there is no commonality in the text converted from images/audio? Usually, the pre-trained models can only convert simple objects/noiseless timbres, which means that the performance of this paper is limited in practical scenarios.

[-] How to solve complex scenarios. Across all datasets in this paper, the scenes are simple and quite detached from reality. How does the performance of  OpenAVS work, especially when dealing with mixed-sound, multi-entity and off-screen scenes?

What’s Making That Sound Right Now? Video-centric Audio-Visual Localization. ICCV2025

**Questions:**

[-] In OpenAVS, multiple pre-trained inferences are concatenated, which results in low inference efficiency. What is the RT of the overall process?

---

### Note · Authors · 2025-11-27

I have read and agree with the venue's withdrawal policy on behalf of myself and my co-authors.